# Comparative analysis of profitability and resource use efficiency between *Penaeus monodon* and *Litopenaeus vannamei* in India

Ubair Nisar[1], Hongzhi Zhang[2], Mahida Navghan[3], Yugui Zhu[1], Yongtong Mu[1]*

**1** Key laboratory of Mariculture (Ministry of Education) College of Fisheries, Ocean University of China, Qingdao, China, **2** Shandong Foreign Trade Vocational College, Qingdao, China, **3** Central Institute of Fisheries Education, Mumbai, India

* ytmu@ouc.edu.cn

## Abstract

The study aimed to highlight the profitability and production function analysis of *Penaeus monodon* and *Litopenaeus vannamei* in intensified shrimp farms in Gujarat (India). Two hundred and twenty (220) shrimp farm households were used to identify (principal component and cluster analyses) 8 clusters of management practices that reflected various scales of production intensity ranging from 0–2999 kg/ha/crop to 9000kg/ha/crop and above for both the species. The Cobb-Douglas production function, which relates production output to several independent input variables, was used to determine productivity. The budgeting analysis for both the species showed that more intensively managed farms performed more than the less intensive farm. Empirical results show feed as most significant input for *Penaeus monodon* and *Litopenaeus vannamei* seed and labor that affected production. Average net returns/ha/year for *Penaeus monodon* was $16313.13 and for *Litopenaeus vannamei* $41640.99. Aquaculture exhibited decreasing returns to scale for both the species and estimates on resource use efficiency revealed that in *Penaeus monodon* the resources were economically utilized and in case of *Litopenaeus vannamei* the output was likely to increase if more of seed and less of labor would have been used. The major constraint for the shrimp farmers was diseases which can be mitigated by optimum stocking densities and proper feed management.

## Introduction

The fisheries sector plays significant role in Indian economy contributing to 0.91% to national GDP and 5.23% to the agricultural GDP [1]. Indian fisheries and aquaculture is an important sector of food production that not only provides livelihood to around 14 million people but also contributes to agricultural exports. Although the shrimp culture has increased during the past decade, the actual potential is still unexploited. The country currently have 176,000 hectares of area under shrimp culture out of which about 91% is under *Litopenaeus vannamei* production, 8% for *Penaeus monodon* and only 1% for *Macrobrachium Rosenberger* [2], Shrimp production can be increased by best utilizing the existing resources through improved

**Data Availability Statement:** All relevant data are within the paper and its Supporting Information files.

**Funding:** Yongtong Mu, acknowledge the financial support of the Ministry of Agriculture and Rural Affairs of Peoples Republic of China (CARS-49).

practices of shrimp culture [3]. Shrimps are called the pinkish gold of the sea because of its increasing demand, great taste and high unit value realization in the export market. It is one of the immersing industry, which significantly contributes to foreign exchange and trade

Gujarat is the fourth largest shrimp producing state in India and emerged as one of the most productive and sustainable shrimp farming state [4]. The total crustaceans exported globally from India in the year 2019 was around 645 million tons worth of 4461 US million dollars (Fig 1). Over the last decade, there has been tremendous increase in the export of crustaceans, majority of which constitutes of frozen seawater shrimps. It is observed in Fig 1 that growth in the crustacean export has increased 4.5 times in terms of quantity and 5.3 times in terms of value since 2001. India also outpaced Indonesia, Thailand, and Ecuador to take the title for most shrimp exports to the U.S. for the fourth straight year [5]. It is a dire necessity of aquaculture to grow in order to provide food for the growing population. However, the growth should be sustainable, and hence responsible for the sustenance in the long run especially for the developing country like India. Studies explain the factors like production (output) intensities and farm sizes as the criterion for the aquaculture sustainability and refer small-scale productions as low input or extensive, while large scale production is referred as intensive [6, 7]. The small-scale production systems mostly use household operated labor and do not rely on hired labors [8]. However, the low intensity farms can be converted into high efficiency production units and managed intensively as large sized by innovations and standardization of procedures [9]. It is argued [10, 11] that low intensity extensive farms tend to be more technically and economically efficient and fetch lower FCR ratio. Similar results were obtained by [12], revealing that though the cost of production per kilogram was highest in semi-intensive culture followed by intensive and extensive culture, the profit per kilogram was highest for extensive culture. Different results obtained by [13–15] revealed that extensive culture is much more profitable than semi-intensive and intensive culture. Shrimp production is mostly dependent on the stocking densities but does not solely influence the production levels [16–18]. High stocking densities and use of high inputs like pelletized feed and medicines characterize the intensive culture. Most of the farmers involved in the culture are urban entrepreneurs with elite businesses and large corporations. The farm owners do not personally take active participation in the management and instead hire managers and technical staff.

In the early 1980's due to high demand and prices of shrimps, the profit margins were very high which lured the investors towards this enterprise resembling a gold rush. However, this expanding industry started encountering the problems since 1988 [19]. Due to farm intensifications, the resource bases started degrading leading to disease outbreak which subsequently led to the drop of shrimp production. Thus, sustainability of shrimp production emerged as a prior concern for long-term viability of the business. The sustainability does not only include

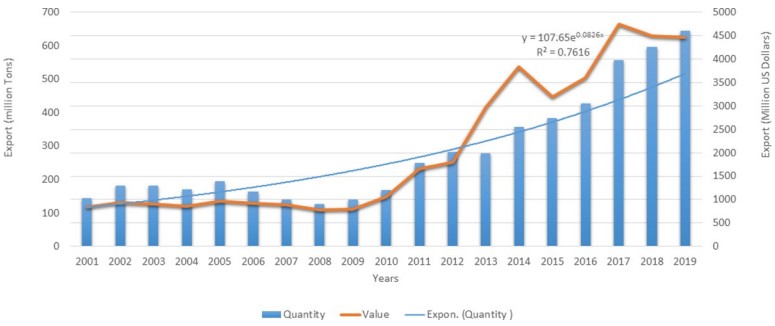

**Fig 1. Performance of Indian crustacean export to the world (2001–2019).**

ecological sustainability but economic sustainability, also determining the capacity of the production system to produce a positive income for the long run.

This study was conducted to assess profitability and resource use efficiency of sample shrimp farms in Gujarat (India) and to determine performances across the levels of intensification.

## Literature review

It is concluded by Engle [20], that production intensification develops "economies of scale" spreading the annual fixed costs over higher production volumes, that ultimately reduces per unit cost of production. By increasing the production cycles and expansion in culture practices results higher production with greater efficiencies and reduced cost of productions.

Penda et al. [21] conducted a study to examine the profitability of fish production in Nigeria, demonstrated that feed, labour and seed were the major components of variable cost sharing 28.10%, 12.76%, and 8.03% in the total cost, respectively. Procurement of feed, labour and seed was the major investment while, pond, pumping machine, harvesting materials shovel, and others were among the fixed assets of production. The elasticity of variables with respect to fish farmers using concrete ponds for feeds, pond size and seed were 0.177, 0.27 and 0.52, respectively. This shows that increasing investment amounts on feeds, fingerlings, and ponds; more production is realized from fish farms.

The study on Resource use of *Litopenaeus vannamei* and *Penaeus monodon* production in Thailand and Vietnam [22], reported that as the production intensity increased, the resources use per metric ton of shrimp reduced. The greater expansion of shrimp ponds with high intensifications leads to lesser use of resources and higher production. The study mentioned the importance of intensification of shrimp farms by stating that in near future to meet the shrimp demands of growing population, the intensification is pivotal. With limited land and water resources, best efficient and productive output can be resulted only by intensification and better management practices.

The study on profitability of intensified shrimp farms in Vietnam and Thailand, revealed that farms with high investments and intensification outclassed with those of less intensifications. Further, the highly intensified shrimp farms in both countries produced greater yields with lower costs per unit of shrimp produced. Higher economic efficiencies were attained in farms with greater intensifications than the lower ones. These efficiencies were accomplished not only by increasing the profit margins but also by reducing the costs [23].

Narayanamoorthy et al. [24], studied the efficiency of shrimp farms in India and suggested that efficiency of a farm relies heavily on the quantities of inputs used. If the stocking density and resources used in the production system are optimum, it leads to healthier economic returns. However, if the resources like feed, stocking density, fertilizers water spread area and available technology are over-utilized it eventually increases the stress and reduces growth rate of shrimps, declining the profit margin.

Shawon et al. [25] in his work estimated the financial profitability of shrimps in coastal areas of Bangladesh, which revealed that culture was economically viable with gross profit margins as high as 59%. Break-even price for shrimps were Tk. 311 per kg while break-even production was found 155 kg per acre. Benefit cost ratio (BCR) was found greater than unity indicating the profitability of the culture with positive net profit margins.

Rasha et al. [26] studied the productivity and resource use efficiency of tiger shrimp revealing that production function for shrimp farming exhibited increasing returns to scale. The major constraints faced by the farmers were high price of inputs (55.20%) followed by insufficient water in dry season (40%) and others.

Radhakrishnan et al. [27] evaluated the input use efficiency of shrimp farming using stochastic product frontier approach. The model was applied to 150 shrimp farmers of India, and the mean efficiency score of 0.95 revealed the high technical efficiency of the farmers. Further inferences revealed that all variables were statistically significant and the small-scale farmers have not improved the efficiency due to least resource utilization and the same can be enhanced by increasing farm investments and intensifications.

## Methodology

### Sampling procedure

The study was carried in the state of Gujarat possessing the largest coastal area and second largest brackish water area in the country. Navsari district was purposively selected for the study as it accounts for largest shrimp farming area in the state [28]. Two blocks namely: Jalalpore and Gandevi with highest shrimp production in the district were selected for the study. From each of the sampled block, two clusters of villages were selected based on area under shrimp culture. A total of 220 shrimp farm households were selected from the district, out of which 100 were for black tiger shrimp farms and 120 for white legged shrimps. The primary data was collected from farmers using multistage stratified simple random sampling and snowball technique and was collected by personal interview method with the help of pre-tested questionnaire especially designed for the study. On the other hand, the secondary data was collected from relevant publications and books. The questionnaire elicited information on the average farm size, stocking density, feeding rates, days of culture, crops per year, equipment's used, average size of shrimps harvested and production input quantities and costs. The farmers were asked to provide the information for the previous production year and the response rates for the survey were 97.50%.

### Analytical technique

In the study, the farms with similar management strategies were grouped and the key component analysis was carried out to classify sets of variables that contributed to the overall variability within the data set. Similar strategies were followed by [29] and [30]. The second stage of the study consist of group observations into clusters of similar characteristics. A cluster analysis was performed to classify groups of farm observations that were similar in terms of key variables than observations in the other clusters such as stocking density, feeding rate, culture days, and total production etc.

### Cost of cultivation

The cost of cultivation of Carp was estimated using the cost concepts defined by Commission of Agricultural Costs and Prices (CACP) [31]. These cost concepts are explained thus;

**Cost $A_1$** = All actual expenses in cash and kind incurred in production by the producer. The items covered in cost A1 are costs on:

I.  value of Post larvae ($)

II.  value of feed ($)

III.  value of medicine ($)

IV.  value of energy (electricity and fuel) ($)

V.  value of hired human labor (permanent and casual) ($)

VI.  land revenue ($)

VII.  interest on working capital (%)

VIII.  communication expenses ($)

IX.  depreciation on fixed capital ($)

X.  repair and maintenance ($)

**Cost A₂** = Cost A1 + Rent paid for leased-in land
**Cost B₁** = Cost A1 + Interest on value of owned capital assets (excluding land)
**Cost B₂** = Cost B1 + Rental value of owned land (net of land revenue) and rent paid for leased-in land
**Cost C₁** = Cost B1 + Imputed value of family labor
**Cost C₂** = Cost B2 + Imputed value of family labor
**Cost C₃** = Cost C2* + 10 per cent of Cost C2* to (on account of managerial functions performed by farmer)

Social aspects of shrimp farming were estimated using descriptive statistics. For each cluster identified, complete enterprise budgeting was performed based on standard techniques of [32]. ANOVA was performed on key parameters like stocking densities, survival rates, yield, feeding rate, FCR, days of culture, number of crops per year and farm size. To arrive at different efficiency measures, analysis was carried out following [33, 34].

## Production function model

The C-D function is expressed as follows:

$$ln\,(Y) = \beta_0 + \sum \beta_1\,ln\,(X_i) + e_i \tag{1}$$

Where, $Y$ denotes output; $Xi$ denotes inputs; $\beta_0$ denotes a constant; $\beta_1$ denotes model coefficients (the elasticities of production) and $e_i$ denotes the random or systematic error.

The empirical Cobb-Douglas production function for this study is expressed as follows:

$$ln\,(Yd) = \beta_0 + \beta_{Fd}ln\,Fd + \beta_{Sd}ln\,Sd + \beta_{Lb}ln\,Lb + \beta_{Md}ln\,Md + \beta_{Ps}ln\,Ps + e_i \tag{2}$$

Where, Yd denotes quantity of shrimp produced (kg/ha/crop): Fd denotes quantity of feed used (kg/ha): Sd denotes post larvae of seeds stocked (Pl/m$^2$): Lb is the labor (man-days/ha): Md is total quantities of medicine used (kg/ha): Ps is the average size of the shrimp farm (ha) and $e_i$ denotes the random or systematic error.

The parameters of investigational significance include:

- Inputs significant to the production process;

- Factor elasticity of each significant input; factor Elasticity ($\beta_i$) measures the marginal change in fish yield from a change in a single input, while other inputs are held constant. This would be obtained from the regression analysis;

- Elasticity of scale ($\varepsilon$) is measured by the percentage change in output with a simultaneous percentage change of equal magnitude in all inputs. The elasticity of scale is the sum of the factor elasticities in the production function

$$\varepsilon = \sum \beta_i\,i = 1\ldots.,n \tag{3}$$

- Allocative efficiency (AE) was determined by calculating the ratio of marginal value product (MVP) and the marginal factor cost (MFC), i.e.

$$AE = MVP/MFC \tag{4}$$

$$\text{And MVP} = \beta_i \frac{\bar{y}}{\bar{x}} P_y \tag{5}$$

Where, MFC = Price per unit of input: $\beta_i$ is regression coefficient of the ith input (i = 1,2,3): $\bar{y}$ is geometric mean of output: $\bar{x}$ is geometric mean of the ith input (i = 1, 2, 3) and Py is price of output.

- The MVP was estimated at the respective geometric mean level and MFC was taken as unit price of the factor. If MVP/MFC equal unity then resource is optimally used. A value of less than unity implies over-utilization of the resource, and of greater than unity under-utilization of the resource.

*Garrett's ranking technique* was used to rank the constraints reported by the farmers on different factors. The shrimp farmers were asked to assign rank to all the constraints faced by them and the outcomes of such rankings were converted into score value thus;

$$\text{Percent position} = 100\,(Rij - 0.5)/Nj \tag{6}$$

where, Rij is the rank given for the ith variable by the jth respondents, and Nj is the number of variable ranked by the jth respondents.

## Results

### Principal component and cluster analyses

Principal component analysis was performed to reduce the dimensionality of the data set and for transforming the larger data into smaller ones with useful information. Seven principal components were found to account for 100% of variability in the data which revealed the internal structure of the data and also explained the variance in Table 1. The variables included in these seven principal components include; culture days, area stocked, total production, mandays of labor use, medicine, total seeds stocked and amount of feed used in production of each crop. The eigenvalues of the first three PCs in the bootstrapped PCA were 2.27, 1.57 and 1.23, respectively, which explained a mean of 72.41% of the total variation in the observed sample.

The farms of both the *L. Vannamei* (white legged shrimp) and *P. Monodon* (Black tiger shrimp) have been categorized as low (0–2999 kg/ha/crop), medium (3000–5999 kg/ha/crop),

**Table 1. Principal components and eigenvalues.**

| Principal Component | Eigen value | Percent Variance | Cumulative Variance |
|---|---|---|---|
| 1 | 2.27 | 32.39 | 32.39 |
| 2 | 1.57 | 22.44 | 54.82 |
| 3 | 1.23 | 17.58 | 72.41 |
| 4 | 0.95 | 13.60 | 86.01 |
| 5 | 0.62 | 8.86 | 94.87 |
| 6 | 0.26 | 3.74 | 98.61 |
| 7 | 0.10 | 1.39 | 100.00 |

**Table 2. Clusters identified for economic analysis.**

| Species | Intensity Category | Yield range (kg/ha/crop) |
|---|---|---|
| *Penaeus monodon* | low | 0–2999 |
| *Penaeus monodon* | Medium | 3000–5999 |
| *Penaeus monodon* | High | 6000–8999 |
| *Penaeus monodon* | Very high | Above 9000 |
| *Litopenaeus vannamei* | low | 0–2999 |
| *Litopenaeus vannamei* | Medium | 3000–5999 |
| *Litopenaeus vannamei* | High | 6000–8999 |
| *Litopenaeus vannamei* | Very high | Above 9000 |

high (6000–8999 kg/ha/crop) and very high (above 9000 kg/ha/crop) as per the yield range as shown in Table 2.

It was observed that in Navsari the mean pond size for *L. Vannamei* and *P. Monodon* was 7937 m² and 4672 m² respectively. Most of the farmers were engaged in the culture of *L. Vannamei*, very few farmers engaged in production of *P. Monodon*. The white legged shrimp was most favored by the farmers because it is more profitable due to its early maturation, high stocking densities, very hardy species and disease resistant, wide tolerance levels and easy acceptance to food. In addition to these advantages, the species are prone to most pathogenic and devastating virus of shrimp (WSSV). The farmers culturing white legged shrimp were constrained by low seed survival, lower growth rate, higher FCR, black gill syndrome (lack of vitamin C), white gut and body cramping (mineral imbalance).

## Production performance

The clusters earlier identified and studied revealed that the farms of *Penaeus monodon* were in the yield range of low and medium clusters, while the farms of *Litopenaeus vannamei* medium, high and very high yielding clusters. With the level of production, the stocking density increased substantially as shown in the Table 3. Yield and feeding rate also increased as the intensity of shrimp production increased. [23] obtained similar results where the yields and feeding intensity increased with the increasing intensification of farms in the *Litopenaeus vannamei* and *Penaeus monodon* culture in Thailand and Vietnam respectively. Higher intensity levels were followed by lower culture days. For *Penaeus monodon*, the medium and low intensity of production were associated with 146 and 158.5 culture days respectively. Similarly for *Litopenaeus vannamei*, the production clusters of very high, high and medium culture

**Table 3. Mean values for key production parameters by categories of intensity/yield levels.**

| | *Penaeus monodon* | | *Litopenaeus vannamei* | | |
|---|---|---|---|---|---|
| | Low | Medium | Medium | High | V. high |
| **Stocking density (Pl/m²)** | 12.00 | 14.00 | 34.13 | 38.56 | 48.50 |
| **Feeding rate (KG/ha/crop)** | 4238.77 | 5167.82 | 6952.63 | 9282.51 | 13858.48 |
| **Culture days** | 158.50 | 146.00 | 154.33 | 141.32 | 127.00 |
| **Yield Kg/ha/crop** | 2563.00 | 3804.00 | 4827.25 | 6271.98 | 10115.68 |
| **FCR** | 1.65 | 1.36 | 1.44 | 1.48 | 1.37 |
| **Harvest weight (shrimps/kg)** | 29.25 | 32.60 | 46.25 | 51.25 | 42.00 |
| **Survival (%)** | 62.40 | 88.57 | 65.40 | 83.30 | 87.00 |
| **Crops/yr.** | 1.00 | 1.00 | 2.00 | 2.00 | 2.00 |

intensities were associated with 127, 141.32 and 154.33 culture days respectively. There was a large variation observed in the stocking densities of both the black tiger shrimp and white legged shrimp. The black tiger was stocked at the rate of 12000 PL per hectare in case of lower intensified to 14000 PL for medium intensified farms. For white legged shrimp, the stocking density varied from 34130 PL/ha for medium intensified farms to 48500 Pl/ha for very high intensity groups.

## Costs and returns in shrimp production

Table 4 shows estimated cost incurred by different levels of intensification in both black tiger and white legged shrimps. It was observed that in *P. monodon*, the total costs incurred in low and medium levels were $8,767.18 and $10,603.06 respectively. For *L. vannamei*, the cost of cultivation of very high cluster farm category was highest ($29,516.70), followed by high farms ($21,781.23) and medium farms ($15,783.93). Feed was the major cost involved in the culture that solely accounted for around 80% of the total variable costs in low and medium intensity cluster of black tiger shrimp. Similarly, for medium, high and very high clusters it accounted for approximately 70% of the total variable costs in the production of white legged shrimps. In aquaculture, feed management is the major factor affecting the water quality and production economics [35, 36]. Correct feeding pattern is important for growth and survival that greatly influence the economic performance of shrimp culture [37]. Variable costs vary with the level of output, so the costs of feed, seed, medicine, energy and labor increased with the intensity of production In the last decade, the intensity of shrimp culture has increased leading to higher stocking densities and greater feed inputs resulting in higher FCR [38, 39]. Total fixed costs per hectare per year also increased with the level of production. The total fixed costs involved the cost of pond and farm building construction, purchase of aerators, feeders, motors, generators, vehicles and others. The additional annual investments like wear and tear (depreciation) and interest on the investment are other components of the fixed cost.

The comparative estimates of different costs incurred in shrimp culture for different levels of production intensities are given in Table 5. The table shows that total cost of production (Cost $C_2$) per hectare of black tiger shrimp is about $9975.52 and $13200.15 on low and medium intensity farms, respectively. Similarly, for white legged shrimps the estimated

**Table 4. Estimated cost of cultivation/ha in shrimp culture (US$/ha).**

| | Penaeus monodon | | Litopenaeus vannamei | | |
|---|---|---|---|---|---|
| | **Low** | **Medium** | **Medium** | **high** | **V. high** |
| **Feed cost** | $4940.55 | $6023.42 | $9024.61 | $12540.61 | $17621.38 |
| **Seed cost** | 120.79 | 164.66 | 2802.73 | 4341.19 | 5652.98 |
| **Medicine and fertilizers cost** | 266.44 | 383.90 | 273.93 | 509.74 | 780.82 |
| **Energy (Electricity and fuel)** | 843.49 | 1017.36 | 777.89 | 878.83 | 1141.81 |
| **Labor** | 35.11 | 43.10 | 51.05 | 85.44 | 139.35 |
| **Interest on working capital** | 263.77 | 324.38 | 549.53 | 780.12 | 1076.79 |
| **Communication costs** | 9.72 | 10.56 | 9.58 | 12.22 | 10.42 |
| **Total Variable costs** | 6,479.87 | 7,967.38 | 13,489.32 | 19,148.15 | 26,423.55 |
| **Depreciation on fixed capital** | 648.20 | 606.50 | 738.92 | 790.90 | 924.80 |
| **Repairs & Maintenance** | 420.32 | 496.03 | 442.25 | 422.55 | 493.42 |
| **Permanent labor** | 501.14 | 535.71 | 357.86 | 413.57 | 528.57 |
| **Interest on fixed capital** | 717.65 | 997.44 | 755.58 | 1006.06 | 1146.36 |
| **Total** | $8,767.18 | $10,603.06 | $15,783.93 | $21,781.23 | $29,516.70 |

**Table 5. Cost concept wise cost of production of shrimps (US$/ha).**

| | Penaeus monodon | | Litopenaeus vannamei | | |
|---|---|---|---|---|---|
| | Low | Medium | Medium | high | V. high |
| A. Cost A1 | 8049.54 | 9605.62 | 15028.36 | 20775.17 | 28370.35 |
| Rent paid for leased in land | 13.89 | 13.89 | 13.89 | 13.89 | 13.89 |
| B. Cost A2 | 8063.43 | 9619.51 | 15042.25 | 20789.06 | 28384.24 |
| Interest on fixed capital | 717.65 | 997.44 | 755.58 | 1006.06 | 1146.36 |
| B. COST B1 | 8767.19 | 10603.07 | 15783.94 | 21781.23 | 29516.71 |
| Rental Value of land+ Rent paid for leased in land | 83.33 | 62.50 | 147.08 | 111.11 | 151.39 |
| C. COST B2 | 8850.52 | 10665.57 | 15931.02 | 21892.34 | 29668.10 |
| Imputed value of family labor | 1125.00 | 2534.58 | 1747.22 | 2204.44 | 3841.32 |
| D. COST C1 | 9892.19 | 13137.65 | 17531.16 | 23985.67 | 33358.03 |
| E. COST C2 | 9975.52 | 13200.15 | 17678.24 | 24096.78 | 33509.42 |
| F. COST C3 | 10973.08 | 14520.17 | 19446.07 | 26506.46 | 36860.36 |

amount is $17678.24, $24096.78 and $33509.42 for medium, high and very high intensity groups respectively. The different measures of costs in shrimp culture viz. costs $A_1$, $A_2$, $B_1$, $B_2$, $C_1$, $C_2$ and $C_3$ are higher for high intensity farms in both *P. monodon* and *L. vannamei*. Cost $C_3$ includes all the possible costs and is considered as the real cost of production in a farm situation. However, rental value of owned land and managerial costs for the farmer can be excluded in a marginal profit situation and Cost $C_1$ can be taken as the standard cost of production, which includes all actual expenses expressed in cash and kind, the rental value of owned capital assets (excluding land) and imputed value of family labor.

It is observed that the cost per kilogram per hectare of shrimps produced decreased with the increase in the production intensity (Table 6). For *Penaeus monodon*, the cost per kilogram per hectare was US$ 4.28 for low intensity level, and later reduced to US$ 3.82 for medium intensity level. Similar pattern was observed for *Litopenaeus vannamei*, the costs decreased from US$ 4.03 for medium intensity to US$ 3.64 for very high intensity of production. The production process of the shrimps followed the economies of scale by sparing in costs and by expanding the culture. Net returns per hectare increased by increasing the level of intensification. As the production increased across the levels of intensification, more volume of output

**Table 6. Returns from cultivation of shrimps on sample farms per hectare (US$).**

| | Penaeus monodon | | | Litopenaeus vannamei | | | |
|---|---|---|---|---|---|---|---|
| | Low | Medium | Average | Medium | High | V. high | Average |
| Yield (Kgs) | 2563.00 | 3804.00 | 3183.50 | 4827.25 | 6271.98 | 10115.68 | 7071.64 |
| Price ($/Kg) | 9.07 | 9.17 | 9.12 | 7.11 | 7.11 | 6.56 | 6.93 |
| Gross Income (GI) | 23253.71 | 34865.80 | 29059.76 | 34343.06 | 44609.96 | 66321.35 | 48424.79 |
| Cost of production ($/Kg) | 4.28 | 3.82 | 4.05 | 4.03 | 4.23 | 3.64 | 3.97 |
| Net Income = GI- Cost C3 | 12280.63 | 20345.63 | 16313.13 | 14896.99 | 18103.50 | 29460.99 | 20820.49 |
| Net income*(per year) | **12280.63** | **20345.63** | **16313.13** | **29793.98** | **36207.00** | **58921.98** | **41640.99** |
| Farm Business Income = GI- CostA2 | 7140.74 | 15640.66 | 11390.70 | 4272.46 | 3045.74 | 9566.76 | 5628.32 |
| Family Labour income = GI-Cost B2 | 14403.19 | 24200.23 | 19301.71 | 18412.04 | 22717.62 | 36653.25 | 25927.64 |
| B:C ratio | 1.12 | 1.40 | 1.26 | 1.53 | 1.37 | 1.60 | 1.50 |

Note

*Since *Litopenaeus vannamei* mature in short duration and 2 crop are taken in a year whereas for *Penaeus monodon* only one crop is taken and hence net income from *Litopenaeus vannamei* crop has been multiplied by 2 to obtain net income per year for comparing with *Penaeus monodon*.

produced resulted in greater gross recipients. For *Penaeus monodon*, the net returns for the lower intensity are $12280.63 and for medium intensity $20345.63 implying the enterprise as a profitable venture. White legged shrimp matures in short duration of time and in that case the culture is done twice a year making the net returns to $29793.98, $36207.00 and $58921.98 for medium, high and very high intensities respectively as shown in Fig 2.

## Factors affecting shrimp production

It is important for economists to understand the inputs that significantly affects the production process and the inputs having higher per unit effect on total production relative to other inputs [40]. The production inputs in this case include; feed (Fd), Seed (Sd), Labor (Lb), Medicine (Md) and Pond size (Ps) of the farmers. Three forms of production function namely, linear, Cobb- Douglas, and Semi log linear were estimated to determine the factors affecting the shrimp farming. Amongst them Cobb-Douglas form of production function was found to be the best fit on both the economic and statistical criteria. The positive production coefficients of the respective inputs in a production function implies that by increasing the intensity of input use, the output can be increased significantly. On the other hand, the negative coefficients suggests that the input should be reduced [40].

The parameters of production function were estimated by step-wise method using SPSS and the results obtained are presented in Table 7 for both the species. The results suggested that the production of *Peneaus monodon* was significantly influenced by the feed (Fd) at 5% level of significance implying that if the feed is increased by 10% percent, the shrimp yield will increase by 2.5%. The model was highly significant (ANOVA gave highly significant F- statistics, with P value significant at 5% level of significance). The $R^2$ was 0.51, implying that 51% of variation in production of *P. monodon* is explained by explanatory variables in the model. Similarly, the production of *L. Vannamei* was influenced by seed (Sd) at 1% level of significance and labor (Lb) at 5% level of significance. [41] and [42] reported similar findings. The $R^2$ was 0.56, implying that 56% of the shrimp yield is explained by explanatory variables in the model.

The models for shrimp production can be expressed as follows:

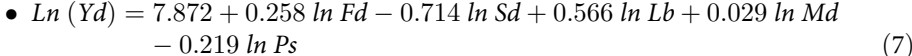

$$Ln\ (Yd) = 7.872 + 0.258\ ln\ Fd - 0.714\ ln\ Sd + 0.566\ ln\ Lb + 0.029\ ln\ Md - 0.219\ ln\ Ps \qquad (7)$$

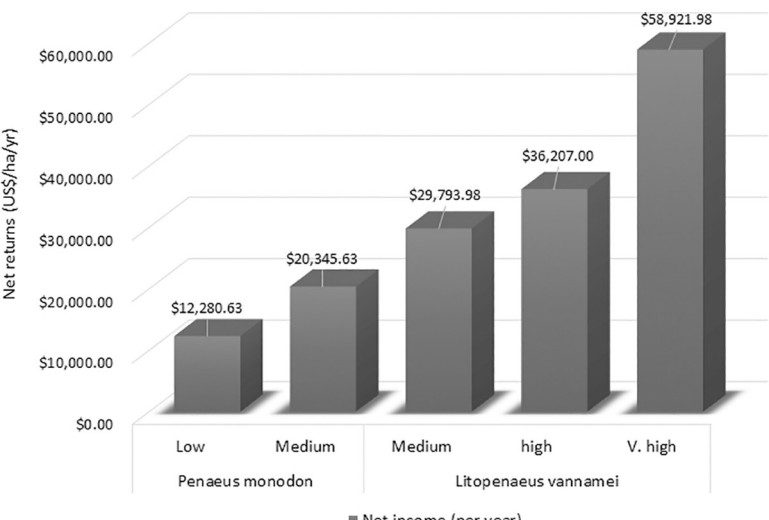

**Fig 2. Net returns (US$/ha/year) in the production of *P. monodon* and *L. vannamei*.**

**Table 7. Cobb-Douglas production function estimation for *Penaeus monodon* and *Litopenaeus vannamei* culture.**

| | *Penaeus monodon* | | *Litopenaeus vannamei* | |
|---|---|---|---|---|
| | *Coefficients* | *Standard Error* | *Coefficients* | *Standard Error* |
| **Const** | 7.872 | 4.913 | 1.667 | 3.066 |
| **Fd** | 0.256** | 0.175 | 0.203 | 0.118 |
| **Sd** | -0.714 | 0.322 | 0.099*** | 0.151 |
| **Lb** | 0.566 | 0.552 | 0.121** | 0.068 |
| **Md** | 0.029 | 0.060 | 0.224 | 0.129 |
| **Ps** | -0.219 | 0.235 | -0.049 | 0.059 |
| **Mean dependent var** | 7.8541 | | 8.7095 | |
| **S.D dependent var** | 0.2202 | | 0.2085 | |
| **Sum squared resid** | 0.6141 | | 1.7608 | |
| **S.E of regression** | 0.1710 | | 0.1956 | |
| **R- squared** | 0.5129 | | 0.5622 | |
| **Adjusted R-squared** | 0.3970 | | 0.4197 | |
| **F value** | 4.4231 | | 2.6131 | |
| **P- value (F)** | 0.0066 | | 0.0367 | |

**indicate significance at 5% level

*** indicate significance at 1% level.

• Fd denotes quantity of feed used (kg/ha).

• Sd denotes post larvae of seeds stocked ($Pl/m^2$).

• Lb is the labor (man-days/ha).

• Md is total quantities of medicine used (kg/ha).

• Ps is the average size of shrimp ponds (ha).

$$\bullet \; Ln\,(Yd) = 1.667 + 0.203\, ln\, Fd + 0.099\, ln\, Sd + 0.121\, ln\, Lb + 0.224\, ln\, Md \\ + 0.049\, ln\, Ps \tag{8}$$

Eq (7) can be used to predict the *P. Monodon* production for farmers, given their production inputs, age and years of experience. Labor was the most powerful explanatory variable with the highest partial output elasticity of 0.566, which means a 10% increase in experience, keeping other inputs constant will increase the yield by 5.66%. Eq (8) predicts the *L vannamei* production and medicine was the most important variable with output elasticity of 0.224 clearly indicating 10% increase in medicines would increase the output by 2.24%. The coefficient of average shrimp pond size was not significant in both the models implying no difference in production between the sizes of different ponds. The level of statistical significance of the estimated production coefficient in both the models (Eqs 7 and 8) are encouraging and there appears to be no problem with multi-collinearity as the Variance inflation factor (VIF) values were lower than 10 (Table 8).

## Returns to scale

The sum of the coefficients in the Cobb-Douglas production function (estimated elasticity function) of any production technology provides returns to scale and is of essential interest given its implications. The results of the study indicates that the *P. Monodon* production in the state has elasticity return to scale of 0.256 ($\Sigma\beta_i$). Since, the estimate is less than one, the production of *P. Monodon* exhibits decreasing returns to scale. This implies that a proportionate

**Table 8. Variance inflation factor (VIF) analysis for multicollinearity.**

| Variables | *Penaeus monodon* VIF | *Litopenaeus vannamei* VIF |
|---|---|---|
| ln Fd | 1.526 | 1.298 |
| ln Sd | 1.636 | 1.071 |
| ln Lb | 1.195 | 1.047 |
| ln Md | 1.250 | 1.073 |
| ln Ps | 4.097 | 2.281 |

*Note- values >10 may indicate a collinearity problem.

increase in inputs will lead to less proportionate increase in output. Similarly, for the *L vannamei* production the elasticity returns to scale is 0.22 ($\Sigma\beta_i$) also exhibiting a decreasing return to scale.

## Resource use efficiency

Resource-use efficiency was estimated for those variables that had significant effect on shrimp production of both the species. It is observed that the efficiency ratio [Marginal Value Product (MVP) to Marginal Factor Cost (MFC)] for *Penaeus monodon* (black tiger shrimp) is greater than unity for feed indicating its under-utilized (Table 9). In the production of *Litopenaeus vannamei* (white legged shrimp), efficiency ratio is greater than unity for use of seed indicating it is underutilized and for labor the ratio is less than unity (over-utilized). Greater than unity values for efficiency ratio in seed and less than unity value for labor exhibits that the output was likely to increase and hence revenue, if more of seed and less of labor would have been used in the shrimp production. In the previous section, the production elasticity of labor has suggested that increase in use of labor will increase shrimp production, however, this increase will not add to the profit of shrimp farmers.

**Table 9. Resource-use efficiency in shrimp farming.**

| | Feed ln Fd | Seed ln Sd | Labor ln Lb |
|---|---|---|---|
| *Penaeus monodon* | | | |
| Geometric mean | 12.60 | - | - |
| Coefficients | 0.26 | - | - |
| Marginal value product (MVP) | 1.82 | - | - |
| Marginal factor cost (MFC) | 1.17 | - | - |
| Efficiency ratio (MVP:MFC) | 1.56 | - | - |
| Decision | Under utilized | - | - |
| Input-use | Increase | | |
| *Litopenaeus vannamei* | | | |
| Geometric mean | - | 12.18 | 8.26 |
| Coefficients | - | 0.10 | 0.12 |
| Marginal value product (MVP) | - | 0.49 | 0.88 |
| Marginal factor cost (MFC) | - | 0.01 | 2.60 |
| Efficiency ratio (MVP:MFC) | - | 49.27 | 0.34 |
| Decision for resource- use | - | Under utilized | Over utilized |
| Input–use | | Increase | Decrease |

## Allocative efficiency of input use

To achieve the most efficient input-use, the value of the marginal value product (MVP) should be equal to its marginal factor cost (MFC) or price [43]. If the MVP of an input is greater than its price, then the profitability can be increased by increasing the level of that input. On the other hand, if the MVP of an input is less than its price then profit can be increased by decreasing that input. In the first regression model of shrimp production (*P. monodon*) as shown in Table 9, feed (Fd) was statistically significant and this input should be increased, since its MVP is greater than MFC. In the second regression model of shrimp production (*Litopenaeus vannamei*), in order to improve the profitability, seed (Sd) should be increased since its MVP is greater than its MFC, whereas the use of labor should be decreased as its MVP is lesser than its MFC.

## Constraints militating against shrimp production among farmers

Farmers were asked to rank their constraints according to their severity. Based on the response of farmers, the Garret score was estimated to find the severity of each constraint and rank was accorded based on Garret score and the results so obtained are presented in Table 10. There are lot of problems which the farmers were facing so the top ten most severe problems have been discussed here. The major problem for both the *P. monodon* and *L. vannamei* was the disease problem. In addition to WSSV, the crop continuously suffered from black gill, white gut problems, running mortality syndrome and shrimp muscle cramping. The disease management requires lot of man hours that increased the labor cost in addition to the high cost of medicines ultimately leading to higher costs of production. The problem could be mitigated by optimum stocking densities, which prevent the overcrowding of shrimps in the ponds. Most of the farmers do not follow the stocking density protocol and overstock the ponds leading to reduction in dissolved oxygen and increase the stress. The second major constraint for the monodon was the high cost of inputs. One of the major input used in the shrimp production is the feed. Feed solely accounted for around 80% of the total variable costs in low and medium intensity cluster. Feed is not only the source of physiological waste but also accounts for 55% to 60% of the variables costs in intensive and around 40% in semi intensive systems [44]. Better-feed management practice will eventually decrease the quantity of feed and ultimately reduce the costs. In case of *L. Vannamei*, the second constraint was availability of skilled labor. Skilled labors have specialized training and skills to perform the operation, so it positively affects the culture practice. They are very useful in taking up the complex physical and mental

**Table 10. Constraints militating against shrimp production among the farmers.**

| Penaeus monodon | | | Litopenaeus vannamei | | |
|---|---|---|---|---|---|
| **Constraints** | **Garret Score** | **Rank** | *Constraints* | **Garret Score** | **Rank** |
| Disease problem | 96.18 | 1 | Disease problem | 94.67 | 1 |
| High cost of input | 64.51 | 2 | Availability of skilled labor | 74.86 | 2 |
| Availability of quality seed | 60.4 | 3 | High cost of input | 63.65 | 3 |
| Availability of skilled labor | 59.26 | 4 | Availability of quality seed | 58.48 | 4 |
| High rate of mortality | 51.24 | 5 | Perishability of produce | 52.14 | 5 |
| Perishability of produce | 42.38 | 6 | High rate of mortality | 44.02 | 6 |
| Pond management | 39.19 | 7 | Price of shrimp | 38.58 | 7 |
| Timely availability | 32.17 | 8 | Credit availability | 31.27 | 8 |
| Price of shrimp | 28.67 | 9 | Governmental schemes | 29.41 | 9 |
| Governmental schemes | 25.54 | 10 | Middle man | 29.18 | 10 |

tasks and carry out quick decision making in any problematic situation. Some other major constraints were availability of quality seed, high rate of mortality, perishability of the produce and others.

## Discussion

The focus of categorizing levels of intensification was to form the farmers with similar characteristics into groups. The categories were made based on the yields of the shrimp farmers and not based on the stocking densities, because stocking density may obscure the effects of intensification on cost efficiencies and profitability. Yields also at the same time are not solely a function of stocking densities but combination of aeration, feed management, medicines and fertilizers and others. [29] noted that profitability of the culture could vary even with similar stocking densities. In this study the stocking densities of *P monodon* for low yielding cluster (12/m$^2$) was similar to medium yielding cluster (14/m$^2$). However, the yield was 48% higher in medium cluster than lower cluster that may be primarily due to 22% higher feeding rates in medium yielding cluster. These results provide evidences from the shrimp culture that supports the results obtained by [29] that use of multivariate tools such as cluster analysis to identify similar sets of management practices as the basis for comparative economic analyses.

The farms within the low yield range had a very extensive level of farming where the water exchange to the farms were completely relied on the tidal flow and used traditional shrimp farming methods. The PL's were purchased from the local hatcheries where the origin of bloodstock is unknown. The farmers were low in finance and did not want to take high risks for enhanced productivity. Ponds were generally harvested according to new moon and full moon pattern. During any disease outbreak, harvesting was done quickly and ponds later chemically treated. Investors in these farms were local residents and usually the farm labor was recruited from the family members or from local communities.

It is observed that harvesting weight of both the species varied considerably with white legged shrimp harvested at higher counts (shrimps/kg) than the black tiger. The possible reason being early maturation in white legged shrimp, which helps in attaining the marketable size in less culture days. The lower harvesting counts for black tiger shrimp was deliberate from the farmers point of view as the species was offered higher and better prices only at lower counts leading to extended culture days of about 160. The number of crops produced per year varied for both the species. The farmers engaged with the culture of *vannamei* were generally involved in two crops per year due to its early maturation, while for *monodon* it was only a single crop per year, as it needs better pre and post stocking management. The survival rates also varied across the clusters with higher intensity clusters having better survival rates. The FCR was found highest in lower yielding intensities for both black tiger shrimps and white legged shrimps and as the intensification of farms increased, the conversion ratio tend to decrease providing better benefit cost ratio.

The study reveals that as the level of intensification increased across the clusters, the profitability also increased. The level of intensification in each cluster increased the profit margins and reduced per unit costs of production by attaining the economy of scale. [45] in his study of intensification of catfish production revealed that with increase in production, the profitability increased with reduction in per unit costs. [46] carried out similar study where the results obtained were contrasting and per unit costs increased with the increase in the production systems. Quite often, the crop failures are blamed on post larvae quality, feed, disease outbreak, and water quality but most often the origin of the failure is poor feed management [47, 48]. So in order to increase the profitability, the farmer should shift from one level to another level of intensification. However, the farmers need more experience, management skills and

capital to shift. In addition to this, the farmers should also possess the risk taking ability, as the shrimp culture is more uncertain for disease outbreak and mortality. The long run profitability of the shrimp farmers in India is affected by the increasing prices of medicines, the cost of hired labors and diseases like WSSV (White Spot Syndrome Virus) which are more prone to *Litopenaeus vannamei*. Land values and construction costs greatly increases the fixed costs, which ultimately hinders the profitability. It is observed that per unit cost of production is affected greatly by the yield but the yield itself is dependent upon the stocking densities, feeding rate, culture days, aeration rate and others. Therefore, yield can be different for same stocking densities per hectare depending on the level and intensities of the other management practices and inputs used.

The parameters of production function were estimated and it is revealed that the quantity of feed used significantly affected the production of black tiger shrimp. The analysis for resource use efficiency depicted feed as an input was economically utilized and results on returns to scale revealed decreasing returns to scale. This implies that a proportionate increase in inputs will lead to less proportionate increase in output. Similarly, for the white legged shrimp, the production was significantly affected by the quantity of seed, and man-days of labor. It was observed that amongst the significant resources, seed was under-utilized and labor over-utilized by the farmers exhibiting that the output was likely to increase and hence revenue, if more of seed and less of labor would have been used in the shrimp production, the production of white legged shrimp also exhibits decreasing returns to scale.

The study revealed that culture of white leg shrimp (*L. vannamei*) and black tiger shrimps (*P. monodon*) are both profitable in the state of Gujarat. However, the white leg shrimp culture is more economically profitable with higher productions mainly due to its early maturation leading to two culture crops per year. Majority of the farmers in the state were involved in white leg shrimp culture due to its high demands and good returns. The farmers indicated the major constraint in culturing the black tiger was the extended culture period and its slow growth. There are also large number of hatcheries producing white leg seeds and limited hatcheries producing the tiger shrimp seed. The white legged shrimp farmers produced good marketable surplus and were economically efficient. In spite of good profitability, the major problem for the farmers was disease outbreak. The diseases like running mortality, muscle cramping, and black gill were more common. The disease outbreak is possibly due to poor feed management. Improvements in feed management will reduce the dependence on fishmeal and will eventually lead to the decrease of nutrient load and reduced requirement for aeration or water exchange in the culture waters.

There are many challenges mitigating against shrimp culture and the farm manager should be skilled enough to manage the more intensive farms. Although the study revealed that in both the species culture, increasing intensification of farms increases the profitability but practically shifting from one level of intensification to another level demands more capital, skills and risk taking ability. There are many case studies in Gujarat where the farmers took up the enterprise without prior knowledge and skills and invested a huge capital in the venture resulting in crop failures and ultimately losses.

Shrimp farming is one of an important productive activity for the population residing near coastal areas. With the increase in the level of intensification and production, the shift takes place from small farmers using their cash crop to sustain families and earn livelihoods to large farmers whose major share of production is exported and provides valuable foreign exchange. In order to increase the production, the farmers are using the strategy of farm intensification that may have environmental, economic and social impacts. In the recent years, aquaculture has grown tremendously due to growing demands (domestic and foreign markets) and promising profits leading to expansion of shrimp farms. This expansion may lead to the

construction in the mangrove areas/ wetlands, which serve as valuable nursery grounds for fish and invertebrates. Intensification leads to higher stocking densities and hence the feeding rates are increased causing overfeeding and water pollution. Many bacterial and viral diseases and the cause in known to the poor management practices leading to biotic and abiotic stress for shrimps mostly affect shrimp farming.

In order to make intensification successful for increasing the production in a sustainable manner, the farmer is to possess sufficient knowledge regarding the culture practices and should have proper management skills to run the shrimp farming successfully in the long run. The farmers should apply proper biosecurity measures in order to prevent intrusions of foreign pathogens for the purpose of disease prevention. The production should be technically and allocatively efficiently produced by better use of the available resources. The farmers should follow Good Management Practices (GMP's) in water and soil quality management, site selection and pond construction, feeding management, seed stocking and harvesting. The effluent treatment systems should be constructed and followed to assist farmers improve the wastewater quality and make their farming practices more sustainable.

## Conclusion

Both *Penaeus monodon and Litopenaeus vannamei* shrimp practices are economically profitable, while white legged shrimp farming was earned more due to its two culture crops in a year and more production compared to black tiger shrimp. It is observed that with increasing intensification of production in both the species the profitability improved resulting in greater yields. With increased intensification, the costs per metric tons of shrimp produced also declined gradually. From the analysis, the main factor influencing the production of black tiger shrimp was feed and for white legged shrimp was seed and labor. The feed as input was considered efficiently used in the production process; however, the profitability of white legged shrimp would have increased if more of seed and less of labor would have been used in the production.

The major constraint mitigating against shrimp farming was the disease problem that can be mitigated by optimum stocking densities and proper feed management. The farmers and farm managers need to be skilled enough to understand the daily nutrient requirement as per the biomass so that the desire of attaining maximum growth does not lead to overfeed and low FCR. If the profitability is to continue in the long run, the economic efficiency and sustainability is to be improved. By employing technically qualified managers on farms can improve the techno-economic efficiency in shrimp farming. It is recommended for extension officers to train more farmers by imparting technology and training wherever necessary and help promote shrimp culture in the area.

## Supporting information

**S1 File.**
(XLSX)

**S2 File.**
(DOCX)

## Acknowledgments

The first author of this article would like to thank the Chinese Scholarship Council (CSC) for the support of my doctoral degree.

## Author Contributions

**Conceptualization:** Ubair Nisar.

**Data curation:** Ubair Nisar.

**Funding acquisition:** Yongtong Mu.

**Investigation:** Yongtong Mu.

**Methodology:** Ubair Nisar, Hongzhi Zhang, Yugui Zhu.

**Supervision:** Yongtong Mu.

**Visualization:** Yongtong Mu.

**Writing – original draft:** Ubair Nisar.

**Writing – review & editing:** Ubair Nisar, Hongzhi Zhang, Mahida Navghan, Yugui Zhu.

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
