## [Editor Report · Decision Letter 0]

11 Jan 2021

PONE-D-20-36703

Comparative Analysis of Profitability and Resource Use Efficiency for Penaeus Monodon vs Litopenaeus Vannamei in India

PLOS ONE

Dear Dr. Yongtong Mu

Thank you for submitting your manuscript to PLOS ONE. After careful consideration, we feel that it has merit but does not fully meet PLOS ONE’s publication criteria as it currently stands. Therefore, we invite you to submit a revised version of the manuscript that addresses the points raised during the review process.

Title: See minor amendments on Title

Abstract: See my inputs in text

Introduction: See my inputs in text

Methodology: In your production function model, age and experience are not/cannot be factors of production but for productivity. Because you estimated a production function, pls expunge these variables and re-run your model, you can use the following: Pond size,  Feed, seed, Medicine and or Capital inputs (depreciated). Please see other inputs in text

Results: See my inputs in text

Discussion: See my inputs in text

Conclusion: See my inputs in text

Please also note the several semantic/language errors many of which were corrected and effect pls.

Your manuscript had a good flow in terms of methodology and discussion if you can effect some of these errors pointed out.

We look forward to receiving your revised manuscript.

Kind regards,

Okoye Benjamin Chukwuemeka

Academic Editor

PLOS ONE

Journal Requirements:

2.) Please include additional information regarding the survey or questionnaire used in the study and ensure that you have provided sufficient details that others could replicate the analyses. For instance, if you developed a questionnaire as part of this study and it is not under a copyright more restrictive than CC-BY, please include a copy, in both the original language and English, as Supporting Information.

3.) We note that you have indicated that data from this study are available upon request. PLOS only allows data to be available upon request if there are legal or ethical restrictions on sharing data publicly. For more information on unacceptable data access restrictions, please see http://journals.plos.org/plosone/s/data-availability#loc-unacceptable-data-access-restrictions.

4.) PLOS requires an ORCID iD for the corresponding author in Editorial Manager on papers submitted after December 6th, 2016. Please ensure that you have an ORCID iD and that it is validated in Editorial Manager. To do this, go to ‘Update my Information’ (in the upper left-hand corner of the main menu), and click on the Fetch/Validate link next to the ORCID field. This will take you to the ORCID site and allow you to create a new iD or authenticate a pre-existing iD in Editorial Manager. Please see the following video for instructions on linking an ORCID iD to your Editorial Manager account: https://www.youtube.com/watch?v=_xcclfuvtxQ

5.) Please include a caption for figure 2.

6.) Please ensure that you refer to Figure 2 in your text as, if accepted, production will need this reference to link the reader to the figure.

Additional Editor Comments:

Title: See minor amendments on Title

Abstract: See my inputs in text

Introduction: See my inputs in text

Methodology: In your production function model, age and experience are not/cannot be factors of production but for productivity. Because you estimated a production function, pls expunge these variables and re-run your mode, you can use the following: Pond size, Feed, seed, Medicine and or Capital inputs (depreciated). Please see other inputs in text

Results: See my inputs in text

Discussion: See my inputs in text

Conclusion: See my inputs in text

Please also note the several semantic/language errors many of which were corrected and effect pls.

Your manuscript had a good flow in terms of methodology and discussion if you can effect some of these errors pointed out.
---

## [Author Response · Author response to Decision Letter 0]

20 Jan 2021

Response to Reviewers

Date: Jan 11 2021 01:14PM

To: "Yongtong Mu" ytmu@ouc.edu.cn

From: "PLOS ONE" plosone@plos.org

Subject: PLOS ONE Decision: Revision required [PONE-D-20-36703]

Ref. No. PONE-D-20-36703 

Title: Comparative Analysis of Profitability and Resource Use Efficiency for Penaeus Monodon vs Litopenaeus Vannamei in India

PLOS ONE

Dear Dr. Yongtong Mu

Thank you for submitting your manuscript to PLOS ONE. After careful consideration, we feel that it has merit but does not fully meet PLOS ONE’s publication criteria as it currently stands. Therefore, we invite you to submit a revised version of the manuscript that addresses the points raised during the review process.

Author’s response: Thank you for your interest in our study. We have made the revision according to your vital suggestions. This will significantly improve the quality of manuscript. The detailed response against each comment is described below sections.

Title: See minor amendments on Title

Author’s response: Thanks for the suggestion. Correction have been made in title as” Comparative Analysis of Profitability and Resource Use Efficiency between Penaeus Monodon and Litopenaeus Vannamei in India”

Abstract: See my inputs in text

Author’s response: Thanks for the advice. Corrections have been made.

Introduction: See my inputs in text

Author’s response: Thanks for the comments. Corrections have been made in whole MS.

Methodology: In your production function model, age and experience are not/cannot be factors of production but for productivity. Because you estimated a production function, pls expunge these variables and re-run your model, you can use the following: Pond size, Feed, seed, Medicine and or Capital inputs (depreciated). Please see other inputs in text

Author’s response: Thanks for the suggestions. Changes have been made and model has been re-run “In the new production function model variables of age and experience has been omitted and the new variable of average shrimp pond size has been added”. 

Results: See my inputs in text

Thanks for the advice. Corrections have been made.

Discussion: See my inputs in text

Author’s response: Thanks for the suggestions. Changes have been made.

Conclusion: See my inputs in text

Author’s response: Thanks for the suggestions. Changes have been made.

Sincerely your’s

Yongtong Mu

---

## [Decision Letter · Decision Letter 1]

22 Mar 2021

PONE-D-20-36703R1

Comparative Analysis of Profitability and Resource Use Efficiency between Penaeus Monodon and Litopenaeus Vannamei in India

PLOS ONE

Dear Dr. Yongtong Mu,

Thank you for submitting your manuscript to PLOS ONE. After careful consideration, we feel that it has merit but does not fully meet PLOS ONE’s publication criteria as it currently stands. Therefore, we invite you to submit a revised version of the manuscript that addresses the points raised during the review process.

You find the evaluations done by the reviewers bellow.

We look forward to receiving your revised manuscript.

Kind regards,

László VASA, PhD

Academic Editor

PLOS ONE

Reviewers' comments:

Reviewer's Responses to Questions

**Comments to the Author**

1. If the authors have adequately addressed your comments raised in a previous round of review and you feel that this manuscript is now acceptable for publication, you may indicate that here to bypass the “Comments to the Author” section, enter your conflict of interest statement in the “Confidential to Editor” section, and submit your "Accept" recommendation.

Reviewer #1: All comments have been addressed

Reviewer #2: All comments have been addressed

2. Is the manuscript technically sound, and do the data support the conclusions?

Reviewer #1: Partly

Reviewer #2: Yes

3. Has the statistical analysis been performed appropriately and rigorously? 

Reviewer #1: (No Response)

Reviewer #2: Yes

4. Have the authors made all data underlying the findings in their manuscript fully available?

Reviewer #1: Yes

Reviewer #2: Yes

5. Is the manuscript presented in an intelligible fashion and written in standard English?

Reviewer #1: Yes

Reviewer #2: Yes

6. Review Comments to the Author

Reviewer #1: The paper is focusing on a rather rarely discussed field so it is absolutely welcome. Authors used appropriate methodology for investigating the pronblem set in the introduction. The overall quality of the article fits PLOS ONE's requirements. However, I have some revision recommendations before publications:

- in fact, there is no literature review chapter in the paper; it is obligartory part of scientific writing so I strongly recommend to write a separated literature review chapter which is analytical, critical and comprehensive enough;

- "Sustainability and intensification" is rather belonging to the discussion chapter, no need for keeping it as separete chapter;

- the figures are of very low quality, those should be reedited/redesigned;

- some parts of the results could be shifted to discussion part, or, these two chapters should be probably merged (Results and discussions)

Reviewer #2: The authors have well revised their manuscript and responded to all comments. I suggest publishing this well-written and technically sound paper.

7. PLOS authors have the option to publish the peer review history of their article (what does this mean?). If published, this will include your full peer review and any attached files.

Reviewer #1: No

Reviewer #2: No

---

## [Author Response · Author response to Decision Letter 1]

1 Apr 2021

Response to Reviewers

Date: Mar 23 2021 

To: "Yongtong Mu" ytmu@ouc.edu.cn

From: "PLOS ONE" plosone@plos.org

Subject: PLOS ONE Decision: Revision required [PONE-D-20-36703R1] - [EMID:855ab397ca70728a]

Ref. No. PONE-D-20-36703R1

Title: Comparative Analysis of Profitability and Resource Use Efficiency for Penaeus Monodon vs Litopenaeus Vannamei in India

PLOS ONE

Dear Dr. Yongtong Mu

Thank you for submitting your manuscript to PLOS ONE. After careful consideration, we feel that it has merit but does not fully meet PLOS ONE’s publication criteria as it currently stands. Therefore, we invite you to submit a revised version of the manuscript that addresses the points raised during the review process. Please submit your revised manuscript by 04.04.2021.

Author’s response: Thank you for your interest in our study. We have made the necessary revisions according to the reviewer’s suggestions. This will significantly improve the quality of manuscript. 

Reviewer Comments to the Author

Reviewer #1: 

The paper is focusing on a rather rarely discussed field so it is absolutely welcome. Authors used appropriate methodology for investigating the problem set in the introduction. The overall quality of the article fits PLOS ONE's requirements. However, I have some revision recommendations before publications:

(1)- in fact, there is no literature review chapter in the paper; it is obligatory part of scientific writing so I strongly recommend to write a separated literature review chapter which is analytical, critical and comprehensive enough

Author’s response: Thanks for the suggestion. We have added the brief literature review chapter in the manuscript as: 

It is concluded by Engle [20], that production intensification develops “economies of scale” spreading the annual fixed costs over higher production volumes, that ultimately reduces per unit cost of production. By increasing the production cycles and expansion in culture practices results higher production with greater efficiencies and reduced cost of productions. 

Penda et al. [21] conducted a study to examine the profitability of fish production in Nigeria, demonstrated that feed, labour and seed were the major components of variable cost sharing 28.10%, 12.76%, and 8.03% in the total cost, respectively. Procurement of feed, labour and seed was the major investment while, pond, pumping machine, harvesting materials shovel, and others were among the fixed assets of production. The elasticity of variables with respect to fish farmers using concrete ponds for feeds, pond size and seed were 0.177, 0.27 and 0.52, respectively. This shows that increasing investment amounts on feeds, fingerlings, and ponds; more production is realized from fish farms.

The study on Resource use of Litopenaeus vannamei and Penaeus monodon production in Thailand and Vietnam [22], reported that as the production intensity increased, the resources use per metric ton of shrimp reduced. The greater expansion of shrimp ponds with high intensifications leads to lesser use of resources and higher production. The study mentioned the importance of intensification of shrimp farms by stating that in near future to meet the shrimp demands of growing population, the intensification is pivotal. With limited land and water resources, best efficient and productive output can be resulted only by intensification and better management practices.

The study on profitability of intensified shrimp farms in Vietnam and Thailand, revealed that farms with high investments and intensification outclassed with those of less intensifications. Further, the highly intensified shrimp farms in both countries produced greater yields with lower costs per unit of shrimp produced. Higher economic efficiencies were attained in farms with greater intensifications than the lower ones. These efficiencies were accomplished not only by increasing the profit margins but also by reducing the costs [23]. 

Narayanamoorthy et al. [24], studied the efficiency of shrimp farms in India and suggested that efficiency of a farm relies heavily on the quantities of inputs used. If the stocking density and resources used in the production system are optimum, it leads to healthier economic returns. However, if the resources like feed, stocking density, fertilizers water spread area and available technology are over-utilized it eventually increases the stress and reduces growth rate of shrimps, declining the profit margin. 

Shawon et al. [25] in his work estimated the financial profitability of shrimps in coastal areas of Bangladesh, which revealed that culture was economically viable with gross profit margins as high as 59%. Break-even price for shrimps were Tk. 311 per kg while break-even production was found 155 kg per acre. Benefit cost ratio (BCR) was found greater than unity indicating the profitability of the culture with positive net profit margins. 

Rasha et al. [26] studied the productivity and resource use efficiency of tiger shrimp revealing that production function for shrimp farming exhibited increasing returns to scale. The major constraints faced by the farmers were high price of inputs (55.20%) followed by insufficient water in dry season (40%) and others. 

Radhakrishnan et al. [27] evaluated the input use efficiency of shrimp farming using stochastic product frontier approach. The model was applied to 150 shrimp farmers of India, and the mean efficiency score of 0.95 revealed the high technical efficiency of the farmers. Further inferences revealed that all variables were statistically significant and the small-scale farmers have not improved the efficiency due to least resource utilization and the same can be enhanced by increasing farm investments and intensifications. 

(2)- "Sustainability and intensification" is rather belonging to the discussion chapter, no need for keeping it as separate chapter;

Author’s response: Thanks for the advice. "Sustainability and intensification" chapter has been merged in the discussion section 

(3)- the figures are of very low quality, those should be reedited/redesigned;

Author’s response: Thanks for the comment. Changes have been made in the figures.

(4)- some parts of the results could be shifted to discussion part, or, these two chapters should be probably merged (Results and discussions)

Author’s response: Thanks for the suggestions. The results have been concised and the explanatory part has been merged in the discussion section. 

Reviewer #2:

 The authors have well revised their manuscript and responded to all comments. I suggest publishing this well-written and technically sound paper.

Author’s response: Thanks for your interest and vital suggestions in improving the quality of our manuscript. 

Sincerely yours

Yongtong Mu

---

## [Decision Letter · Decision Letter 2]

13 Apr 2021

Comparative Analysis of Profitability and Resource Use Efficiency between Penaeus Monodon and Litopenaeus Vannamei in India

PONE-D-20-36703R2

Dear Dr. Yongtong Mu,

We’re pleased to inform you that your manuscript has been judged scientifically suitable for publication and will be formally accepted for publication once it meets all outstanding technical requirements.

Kind regards,

László VASA, PhD

Academic Editor

PLOS ONE

Additional Editor Comments (optional):

Reviewers' comments:

Reviewer's Responses to Questions

**Comments to the Author**

1. If the authors have adequately addressed your comments raised in a previous round of review and you feel that this manuscript is now acceptable for publication, you may indicate that here to bypass the “Comments to the Author” section, enter your conflict of interest statement in the “Confidential to Editor” section, and submit your "Accept" recommendation.

Reviewer #1: All comments have been addressed

Reviewer #2: All comments have been addressed

2. Is the manuscript technically sound, and do the data support the conclusions?

Reviewer #1: Yes

Reviewer #2: Yes

3. Has the statistical analysis been performed appropriately and rigorously? 

Reviewer #1: Yes

Reviewer #2: Yes

4. Have the authors made all data underlying the findings in their manuscript fully available?

Reviewer #1: Yes

Reviewer #2: Yes

5. Is the manuscript presented in an intelligible fashion and written in standard English?

Reviewer #1: Yes

Reviewer #2: Yes

6. Review Comments to the Author

Reviewer #1: Authors made the required improvements, I accept it for publication. Literature review was improved, and also other relevant parts, so the paper is scientifically sound and comprehensive.

Reviewer #2: The authors have well addressed all the previous comments and now the paper is ready for publication.

7. PLOS authors have the option to publish the peer review history of their article (what does this mean?). If published, this will include your full peer review and any attached files.

Reviewer #1: No

Reviewer #2: No

---

## [Editor Report · Acceptance letter]

15 Apr 2021

PONE-D-20-36703R2 

Comparative Analysis of Profitability and Resource Use Efficiency between *Penaeus Monodon* and *Litopenaeus Vannamei* in India 

Dear Dr. Mu:

I'm pleased to inform you that your manuscript has been deemed suitable for publication in PLOS ONE. Congratulations! Your manuscript is now with our production department. 

Kind regards, 

on behalf of

Prof. Dr. László VASA 

Academic Editor

PLOS ONE